# TAMING ENCODER FOR ZERO FINE-TUNING IMAGE CUSTOMIZATION WITH TEXT-TO-IMAGE DIFFUSION MODELS

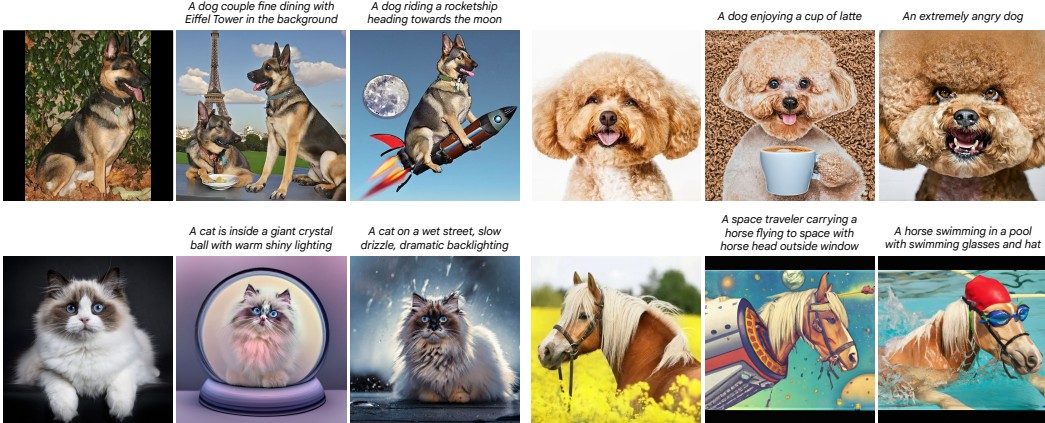

Figure 1: Given *one* reference image and a text prompt, our method generates an image containing the same object seamlessly immersed with novel concepts described by the text in a ***single forward pass***, *e.g., puffed up fur for angry dog*.

## ABSTRACT

This paper presents a novel approach for creating customized images of objects as per user specifications. Unlike previous methods that often involve time-consuming optimizations, typically following a per-object optimization approach, our method is built upon a comprehensive framework designed to expedite the process. Our framework employs an encoder to capture the essential high-level characteristics of objects, generating an object-specific embedding through a single feed-forward pass. This acquired object embedding is subsequently utilized by a text-to-image synthesis model for image generation. To seamlessly integrate the object-aware embedding space into a well-established text-to-image model within the same generation context, we explore various network architectures and training strategies. Furthermore, we introduce a straightforward yet highly effective regularized joint training approach that incorporates an object identity preservation loss. In addition to this, we propose a caption generation scheme that plays a crucial role in ensuring the faithful representation of object-specific embeddings throughout the image generation process. This approach enables users to maintain control over the process and provides them with editing capabilities.

## 1 INTRODUCTION

Text-to-image synthesis Saharia et al. (2022); Ramesh et al. (2021); Yu et al. (2022); Rombach et al. (2022); Sauer et al. (2023); Chen et al. (2022a) has gained increasing attention and experienced rapid development with recent advances of GANs Goodfellow et al. (2020); Karras et al. (2019); Brock et al. (2018); Karras et al. (2018; 2020; 2021) and diffusion models Ho et al. (2020); Song et al. (2020). With sophisticated designs and enormous amount of training data Schuhmann et al.

(2022); Radford et al. (2021), images with unprecedented quality and diversity can be generated through conditioning on free-form texts provided by users. Apart from generic objects, an intriguing question would be *whether it is possible to synthesize images capturing an object specified by users*. Generating a particular object requires the understanding of its high-level concept, which is intricate, if not impossible, when the desired object is not contained in the training set. To remedy the domain gap, existing methods Gal et al. (2022); Ruiz et al. (2023); Brooks et al. (2023); Kumari et al. (2023); Mokady et al. (2022) generally adopt a fine-tuning paradigm, where a text prompt representing the object and a pre-trained synthesis model are jointly optimized using multiple images of the object provided by users. The optimized text prompt is then combined with natural language descriptions to generate outputs containing the objects with various content and styles.

Training a model for each object is infeasible to scale up for practical uses. In particular, as object-specific fine-tuning is required, the aforementioned paradigm is unable to produce fast adaptation to arbitrary objects. The applicability of such methods is further limited by the intractable model storage cost, which increases with the number of objects considered. Furthermore, these methods usually require multiple images of the same object, which is not always available in reality.

The main focus of this paper is *learning a single general model that is able to compose a new scene around given objects yet without the need of per-object optimization, using as few as one image*. This is an unexplored direction in spite of its wide applicability. In this work, we introduce a general framework as the first step towards this goal. Unlike existing works Ruiz et al. (2023); Kumari et al. (2023); Mokady et al. (2022) that discretely align a target object with a unique prompt through iterative optimization, we aim to continuously project object embedding and text-to-image generation embedding into an unified semantic space.

Despite its apparent simplicity, two non-trivial challenges emerge: 1) the adaptation of text-to-image models to object embeddings, and 2) the development of training data that enhances personalization while preserving the editing capabilities of pre-trained models.

To address these challenges, we introduce additional attention modules into a text-to-image network when provided with object embeddings as an extra input. This augmented network undergoes fine-tuning to incorporate the object embedding as a conditioning factor. However, direct fine-tuning with the object embedding leads to a decline in editing capabilities. To circumvent this issue, we propose a novel approach termed the *regularized joint training scheme* coupled with *cross-reference regularization*. This method enables us to maintain editing flexibility while effectively integrating object-conditioning into the model. Furthermore, we present a method for enhancing the training dataset by generating captions. This approach enhances data precision, ultimately yielding improvements in output quality, diversity of appearances, and object accuracy.

Given the increasing importance of personalized text-to-image synthesis in content creation, the need for an efficient algorithm cannot be overstated. In this study, we initiate efforts in this direction, showcasing a method to streamline the optimization process without compromising performance. Our resulting model is both straightforward and capable of generating personalized images in a single feed-forward pass, resulting in reduced computational and storage expenses.

## 2 RELATED WORK

**Text-to-Image Synthesis.** Approaches for text-to-image synthesis Saharia et al. (2022); Ramesh et al. (2021); Yu et al. (2022); Rombach et al. (2022); Sauer et al. (2023); Kawar et al. (2023); Chang et al. (2023); Ramesh et al. (2021; 2022); Yu et al. (2022); Pan et al. (2023); Sheynin et al. (2022); Nichol et al. (2022;?); Wu et al. (2022); Liao et al. (2022) can be divided into three main categories. The vector-quantized approach Yu et al. (2022); Chang et al. (2023); Ding et al. (2022) first learns a discrete codebook through training an autoencoder. After training, earlier works Yu et al. (2022); Ding et al. (2022) adopt an autoregressive transformer to predict the tokens sequentially. The predicted tokens are then passed to the decoder to generate the output image. To reduce computational cost for high-resolution images, bidirectional transformers Chang et al. (2023) are introduced to predict the tokens all at once and iteratively refine them. In this case, the computational time is reduced by eliminating the sequential operations. Diffusion models Rombach et al. (2022); Saharia et al. (2022); Sheynin et al. (2022); Nichol et al. (2022) synthesize images through iterative denoising. Starting from a standard Gaussian noise, a UNet is usually adopted to denoise the intermediate outputs, conditioning on the text prompt, to produce a less noisy outputs. The fi-

nal output is obtained through repeating the denoising process. Recently, StyleGAN-T Sauer et al. (2023) demonstrates the capability of the GAN framework Tao et al. (2022); Huang et al. (2022); Liao et al. (2022) in text-to-image synthesis by modifying the StyleGAN Karras et al. (2018; 2020; 2021) architecture, allowing text conditioning. The aforementioned approaches achieve compelling performance with the presence of large-scale text-image datasets Schuhmann et al. (2022). In this work, we follow the diffusion model paradigm. In particular, we adopt Imagen Saharia et al. (2022) as our backbone network.

**Personalized Synthesis.** Most existing works for personalized synthesis adopt a pre-trained synthesis network and perform test-time training for each object. For instance, MyStyle Nitzan et al. (2022) adopts a pre-trained StyleGAN for personalized face generation. For each identity, it optimizes the latent code as well as fine-tuning the pre-trained network. Once trained, it can be used to synthesize images of the target identity. This paradigm is also seen in the task of text-to-image synthesis Gal et al. (2022); Ruiz et al. (2023); Brooks et al. (2023); Kumari et al. (2023); Mokady et al. (2022). Given a pre-trained text-to-image synthesis model and multiple images containing the target object, a text prompt representing the objects are optimized and the network is optionally fine-tuned to further adapt to the target object. After training, the optimized object text prompt can be combined with natural language descriptions to generate diverse outputs. With test-time optimization required, the aforementioned approach usually requires minutes to hours for each object. In addition, the model storage cost increases with the number of objects to be handled. As a result, their scalability and practicality are essentially limited. In this work, we focus on bypassing the lengthy optimization, enhancing the efficiency of personalized text-to-image synthesis.

## 3 APPROACH

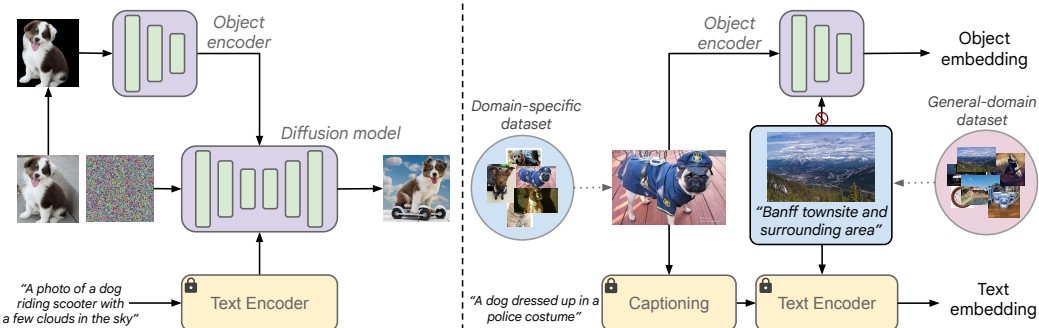

Figure 2: **Framework Overview.** **(Left)** Given a reference image, the background is removed, and an encoder is used extract embedding from the filtered image. The object embedding is then used along with the text embedding for subsequent generation. **(Right)** Our triplet preparation scheme. For domain-specific dataset, which generally does not caption, we apply a captioning model to obtain corresponding caption. Note that the object embedding is set to a null embedding for general-domain images.

### 3.1 OVERVIEW

**Framework.** In Fig. 2, we illustrate our framework's structure, which comprises two primary elements. The initial component involves a text-to-image synthesis network, as described in the works Saharia et al. (2022); Rombach et al. (2022). To facilitate text-driven synthesis, we employ a pre-trained model. The second component consists of an image encoder, employed to encode diverse objects into object embeddings. These embeddings serve as the secondary conditioning factor for object-specific generation within the synthesis network. To accommodate object embeddings not inherent in the pre-trained model, we enhance the network by incorporating cross-attention modules.

Specifically, let $\mathbf{x}$ be an image containing the object of interest and $\mathbf{c}$ be text caption describing a desired output image. The image object encoder $\mathcal{I}$ and text encoder $\mathcal{T}$ are used to compute the object embedding $\mathcal{I}(\mathbf{x})$ and text embedding $\mathcal{T}(\mathbf{c})$, respectively. The embeddings are then passed to the augmented text-to-image diffusion model, such as Stable Diffusion Rombach et al. (2022) and Imagen Saharia et al. (2022), to generate the final output. It's essential to note that our framework

maintains a generic nature and isn't limited to any specific architecture. For this study, we opt to utilize Imagen Saharia et al. (2022) as the synthesis network.

In contrast to the majority of prior approaches, which involve iterative optimization for each individual object, our novel framework achieves image generation for an object with a single forward pass. This streamlined approach significantly diminishes the computational burden and storage requirements typically associated with per-object optimization.

**Encoder.** The object encoder, denoted as $\mathcal{I}$, plays a pivotal role in grasping the essence of an object within our framework. In principle, one could employ networks of various architectures as the encoder, but we have observed substantial variations in performance associated with different network choices. Our working hypothesis is that, in order to capture abstract concepts effectively, the encoder should undergo training with two key factors: 1) an extensive dataset and 2) an objective function that links concrete objects with abstract descriptions. Our hypothesis gains empirical support through our investigations later in Sec. 5.

## 3.2 PRELIMINARY

**Imagen.** Conditioned on text embeddings $\mathbf{c}$, Imagen is trained with a denoising objective

$$\mathbb{E}_{\mathbf{x},\mathbf{c},\epsilon,t}\left[||\epsilon_\theta(\mathbf{x}_t,t,\mathbf{c})-\epsilon||_2^2\right],\tag{1}$$

where $\mathbf{x}_t$ is a noisy version of the groundtruth image, $\epsilon \sim \mathcal{N}(\mathbf{0},\mathbf{I})$ denotes standard Gaussian, and $t$ is the timestep.

Classifier-free guidance Ho & Salimans (2022) is a commonly used technique for improving diffusion model sample quality. It efficiently train one model for both conditional and unconditional objectives, achieved by randomly dropping the condition $\mathbf{c}$. During sampling, the intermediate predictions are adjusted based on the conditional and unconditional outputs as:

$$\hat{\epsilon}_\theta(\mathbf{x}_t,\mathbf{c})=w\epsilon_\theta(\mathbf{x}_t,\mathbf{c})+(1-w)\epsilon_\theta(\mathbf{x}_t,\mathbf{c}_\emptyset),\tag{2}$$

where $w$ is the guidance weight and $\mathbf{c}_\emptyset$ is the embedding of an empty text string. Intuitively, the guidance effect increases with $w$.

Rather than generating images directly at the final resolution of $1024 \times 1024$, Imagen employs a cascaded approach. Initially, it produces a $64 \times 64$ image, followed by the utilization of two text-conditioning super-resolution models to scale up the image to $256 \times 256$ and subsequently to $1024 \times 1024$.

## 3.3 TRIPLET PREPARATION

This section discusses the keys in preparing triplets for our network fine-tuning. Specifically, we first discuss our caption generation scheme, followed by our object embedding generation.

**Captioning with PaLI.** One could improve personalization by training with objects of the same category. For example, the identity preservation of a dog is improved when datasets containing animals are included during training. However, such datasets usually contain no text captions, prohibiting domain-specific fine-tuning.

In this work, we propose to apply a language-image model, PaLI Chen et al. (2022b), on the images to generate descriptive captions. Let $f_c$ be a captioning model, we apply it to the image $\mathbf{x}$ to obtain the *coarse caption* $\mathbf{c}_c = f_c(\mathbf{x})$.

In addition to the general description provided by PaLI, we further incorporate concrete attributes (*e.g.*, face attributes) if they are available. Let $f_a$ be the attribute classification network, our *fine caption* $\mathbf{c}_f = f_a(\mathbf{x})$ is generated by applying $f_a$ to the image. The two captions are then concatenated as the final caption $\mathbf{c}$. Our approach generates captions with both abstract and concrete description, circumventing the loss of text-conditioning property.

**Background-Masked Object Embedding.** Intuitively, the object embedding should depend only on the object, and should be agnostic to the background. Therefore, to better disentangle the object from the input image, we apply a binary mask to remove the background. Let $f_m$ be the mask generation function, we have $\mathbf{o} = f_o(\mathbf{x} \otimes \mathbf{m})$, where $\otimes$ denotes pointwise multiplication, and

$\mathbf{m} = f_m(\mathbf{x})$ denotes the binary mask. In such a way, the object embedding depends only on the object, possessing higher object specificity.

### 3.4 REGULARIZED JOINT TRAINING

Pretraining a model on just a few images can sometimes cause overfitting and the risk of forgetting previously acquired skills, as highlighted by He et al. (2019). Nevertheless, our framework addresses a vast array of objects within similar categories, allowing for joint training with traditional text-to-image data. Additionally, personalization can be enhanced by simultaneously training on large-scale text-image datasets and domain-specific datasets. However, blending these two datasets directly can result in a loss of text-conditioning influence, with the object embedding taking precedence in the generation process. Effectively conditioning the network on the additional object embedding without sacrificing text-to-image synthesis capability remains a formidable challenge. To tackle this issue, we propose a "regularized joint training scheme" designed to 1) prevent the object embedding from dominating and 2) improve the fidelity of object representation.

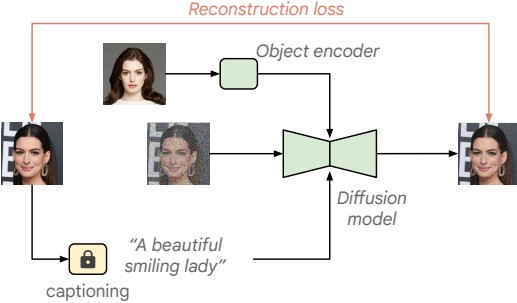

Figure 3: **Cross-reference regularization.** Since object embedding is agnostic to appearances, we use another image of the same identity to compute object embedding, encouraging disentanglement of identity information.

**Cross-reference regularization.** Motivated by the fact that the object embedding is shared across images of the same object, we propose the cross-reference regularization to encourage disentanglement of identity information from the object embedding (Fig. 3). Given an additional image $\bar{\mathbf{x}}$ capturing the same object, we randomly replace the object embedding $\bar{\mathbf{o}}$ by that computed from $\bar{\mathbf{x}}$:

$$\mathbf{z} = \begin{cases} (\mathbf{x}, \bar{\mathbf{o}}, \mathbf{t}) & \text{if } p < \omega, \\ (\mathbf{x}, \mathbf{o}, \mathbf{t}) & \text{otherwise,} \end{cases} \tag{3}$$

where $p \sim \mathcal{U}(0, 1)$ and $\omega$ is a pre-determined threshold. Here $\mathbf{z}$ represents the training triplet. Note that the regularization is applied only for domain-specific images. Then, the synthesis network learns to distill object-specific information from the object embedding, essentially removing image-specific clues. We find that this significantly improve the object fidelity.

**Object-Embedding Dropping.** In practice, one could apply the object encoder to the images in the general-domain dataset. However, we observe that the generation process is dominated by the object embedding. As a result, the trained network generates outputs solely based on the object embedding, neglecting the text conditions. To avoid this, we impose an implicit regularization to reduce the reliance on the object embedding, especially for general-domain images. Let $\mathbf{z}$ be the triplet during training, the object embedding is set to a null embedding for general domain images:

$$\mathbf{z} = \begin{cases} (\mathbf{x}, \mathbf{o}, \mathbf{t}) & \text{if } \mathbf{x} \text{ is domain-specific,} \\ (\mathbf{x}, \phi, \mathbf{t}) & \text{otherwise,} \end{cases} \tag{4}$$

where $\phi$ denotes a fixed null embedding. In this case, the network learns to leverage object embedding for domain-specific objects, while keeping the capability of text-conditioning through reconstructing general-domain images solely with text embedding.

**Whole-Network Tuning.** In the test-time optimization paradigm, a recent work Kumari et al. (2023) discovers that fine-tuning only the attention modules lead to a comparable results. However, it is

observed that training only the extra attention module while fixing the pre-trained backbone results in inferior performance in our framework. In particular, when training only the attention layer, the network is inferior in preserving the identity of the object despite being able to synthesizes corresponding context based on the text captions. Therefore, we unfreeze all the network weights and train them jointly.

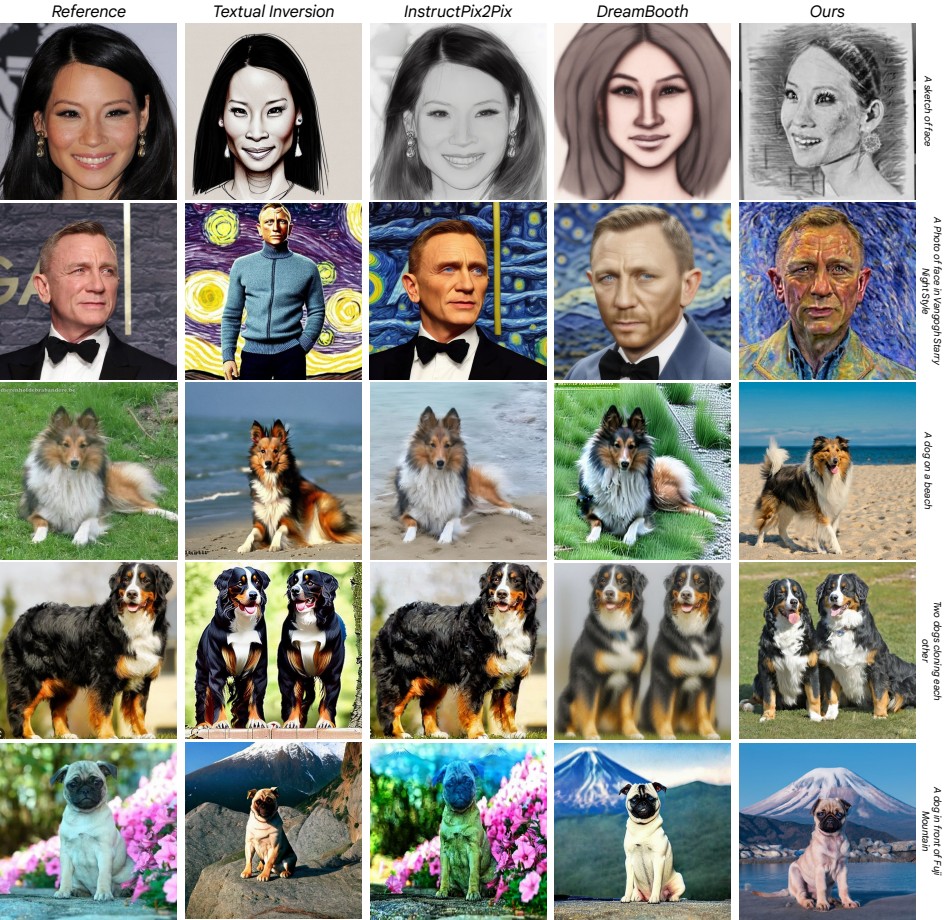

Figure 4: **Qualitative Comparison.** For human face styling, existing works sometimes either struggle to generate styles faithful to the texts or fail to preserve identity. For animals, existing works often overfit to the input image, generating outputs with limited diversity. Note that for Textual Inversion and DreamBooth, five images are used as input for human faces and one image is used for animals.

## 4 EXPERIMENTS

**Settings.** We adopt pre-trained Imagen Saharia et al. (2022) as the text-to-image network. The text embeddings are computed by T5-XXL Raffel et al. (2020). We use CLIP Radford et al. (2021) image encoder to generate the object embeddings, and insert additional attention modules to Imagen to accommodate the object embeddings. The entire network except the object encoder is jointly trained with the same objective as used in training Imagen. We evaluate our method on two categories. For human faces, we incorporate the CelebA Liu et al. (2015) into our internal large-scale text-image dataset and train jointly on the combined datasets. For animals, we adopt LSUN-dog Yu et al. (2015) for fine-tuning. The models are fine-tuned for around two days with 64 TPU-v4 chips. More details on the architecture and training settings are discussed in the supplementary material. We compare our performance with three existing state of the arts, namely Textual Inversion Gal et al. (2022), DreamBooth Ruiz et al. (2023), and InstructPix2Pix Brooks et al. (2023). We use their publicly released code, for Dreambooth, we use the implementation in diffuser for comparison.

**Comparison.** In Fig. 4, we present a comparison between our approach and existing methods. Initially, we showcase the style-editing capacity of our method using human faces as an example. In some cases, existing methods encounter challenges in generating styles that faithfully correspond to the provided texts or in preserving critical identity details. For instance, when given the text "A sketch of a face," all methods except ours struggle to produce a sketch-like image. Notably, Textual Inversion and DreamBooth require five images as input, whereas our method accomplishes the task with just one image.

Subsequently, we illustrate our capability to synthesize diverse contexts involving animals. When given only one image as input, existing methods often exhibit a tendency to overly conform to the input image in terms of pose and gesture, resulting in limited diversity in their generated outputs. In contrast, our method robust to the number of input images and is able to generate images with diverse poses and context. For example, only our method is able to generate two identical dogs with different poses. Furthermore, although our method is not trained on cat-specific and horse-specific datasets, it is generalizable to a wider class (*i.e.*, animals), as shown in Fig. 1.

In addition to generalizability and quality, our method possesses greater efficiency. Specifically, while the training and storage costs of the methods in comparison increases linearly with the number of objects, our method does not require any per-object training, and hence the costs remain constant. The aforementioned strengths of our method essentially ease the use of personalized synthesis, unleashing human creativity.

We further compare with baselines using quantitative metrics, including object similarity (OSim.), caption similarity (TSim.) and Kernel Inception Distance (KID) Bińkowski et al. (2018). Object similarity measures the distance between input subject and personalized output using the pretrained CLIP image encoder. Caption similarity measures the distance between personalized output and the prompt using the pretrained CLIP text encoder. KID is useful in evaluating how close our personalized output distribution is to the style given by prompts. Table 1 consistently demonstrates that the proposed approach performs better in preserving object identity and matching user prompts with a single input.

Table 1: Quantitative comparison. (TI-$n$: Textual Inversion with $n$ image. DB: DreamBooth. Pix2Pix: InstructPix2Pix. SD: Stable Diffusion)

| Methods | TI-1 | TI-5 | DB-1 | DB-5 | Pix2Pix | Ours(SD) | Ours(Imagen) |
|---|---|---|---|---|---|---|---|
| OSim.↑ | 0.23 | 0.31 | 0.33 | 0.34 | 0.37 | 0.41 | **0.46** |
| TSim.↑ | 0.18 | 0.20 | 0.28 | 0.31 | 0.36 | 0.33 | **0.35** |
| KID↓ | 20.34 | 17.31 | 24.57 | 16.89 | 15.08 | 13.57 | **13.23** |

## 5 ABLATION STUDIES

**Choice of Encoder.** We train a diffusion model on faces with object embedding as the sole condition to demonstrate the importance of choosing an appropriate encoder. As depicted in Fig. 5, when using VGG19 Simonyan & Zisserman (2015), which is trained for classification, as the encoder, the object embedding is unable to capture high-level concept of a face, thus generating random faces. In contrast, when CLIP is adopted, the network is able to generate faces with the same identity. Moreover, through relating abstract concept and concrete objects during the training of CLIP, the object embedding is agnostic to image-specific clues, leading to outputs with variations. This demonstrates the importance of the encoder. More sophisticated designs of the encoder are left as our future work.

**Cross-reference regularization.** Since CLIP is not dedicated for identity preservation, we observe that directly using CLIP embedding leads to imperfect identity preservation and excessive retention of image clues. From Fig. 6 we see that without regularization, the synthesized images often overfit to the fine details (*e.g.*, hair styles) or insufficiently capture the identity. Our proposed regularization scheme alleviates the issues by swapping the object embeddings from the same object, enforcing identity disentanglement. As a result, the generated images possess greater diversity while preserving the identity.

**Whole-Network Tuning.** For methods involving test-time optimization, it is shown that fine-tuning only the attention modules leads to improved training efficiency without compromising output qual-

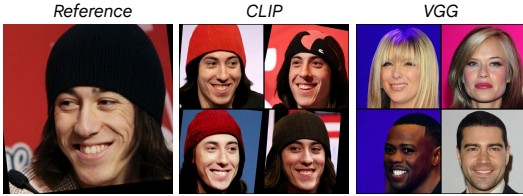

Figure 5: The CLIP image encoder preserves identity while allowing appearance variations. In contrast, VGG generates random faces.

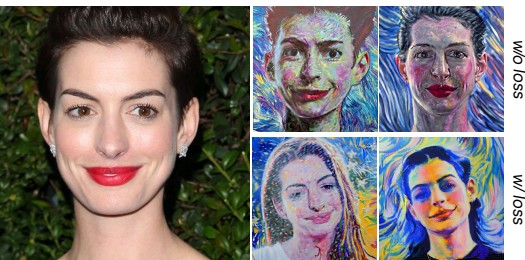

Figure 6: Cross-reference regularization disentangles object concept from image-specific clues, leading to outputs with better *identity preservation* and *appearance diversity*.

ity Kumari et al. (2023). However, we find in our framework that training only the added attention modules results in inferior performance. Specifically, as shown in Fig. 7, while fine-tuning only the attention layer retains the text-conditioning capability, identity cannot be preserved. In contrast, both output quality and identity preservation are improved when all weights are jointly trained. Our conjecture is that, unlike previous works that are confined to limited objects, our network is trained to generalized to unseen objects, and whole-network tuning enables more effective exploitation of the object embedding, improving generalization ability.

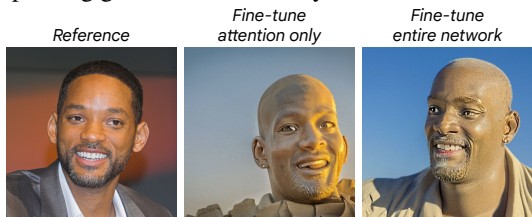

Figure 7: Whole-network tuning improves object embedding exploitation.

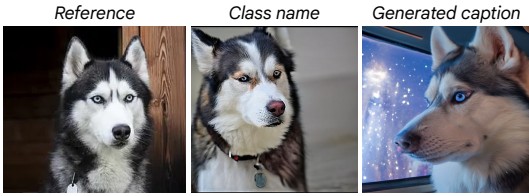

Figure 8: Natural captions enables the model to produce accurate and faithful results.

**Caption Generation.** Our caption generation scheme provides diverse captions on domain-specific datasets for our joint training scheme. When compared to the naïve approach of simply setting the class name (*e.g.*, dog) as the caption, our autocaptioning scheme leads to significantly better performance. For example, as shown in Fig. 8, when only trained with class name, while the network is able to generate the same objects, it is inferior in generating content faithful to the text captions. This is due to the domain gap between the captions in the general-domain dataset and the domain-specific dataset. In contrast, our training scheme remedies the domain gap between the two datasets by synthesizing descriptive captions. As a result, the trained network is able to preserve object

identity as well as generating contexts according to the given text caption. For example, in the first example, while the network trained with only class name is unable to generate outputs related to "spacecraft" and "night", our method generates faithful results with high fidelity to the text caption.

**Distribution Mixing.** As discussed in the previous section, it is possible to incorporate domain-specific datasets into large-scale text-image datasets for improving personalization. As shown in Fig. 9, on the one hand, when we fine-tune the network on the general-domain dataset only (second row), the network is able to produce an output conforming to the text (*i.e.*, a cartoon face), but fails in resembling the identity. On the other hand, when trained only on domain-specific dataset, the network ignores the texts and produces a natural face that resembles the reference identity. The gradual transition shows that it is essential to balance the ratio of the two datasets.

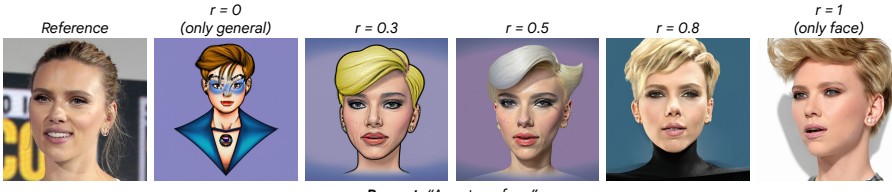

Figure 9: Proper dataset mixing ratio helps identity preservation while alleviating forgetting issues.

## 6  LIMITATION AND SOCIETAL IMPACT

We observe that the outputs of our method often contain defects when the corresponding details are not presented in the original image. For example, in Fig. 10, when the right eye of the dog is not shown in the input image, the network either ignores (output 1) or hallucinates (output 2) the right eye, and hence incoherence is observed in the outputs. In the future, we will extend the framework so that multiple images are taken as inputs, improving the robustness. This work can inherit the bias originated from training data, e.g, CelebA, which brings a consequent bias toward images of attractive people who are mostly in age range of twenty to forty years old. It may also contain only few images of certain group of race, which can potentially lead to misleading content creation and stereotyping propagation. Single image personalization may increase the ability to forge convincing images of non-public individuals. To prevent this, future efforts should be devoted to both the generative side (*e.g.*, cleaning training data) and discriminative side (*e.g.*, forgery detection).

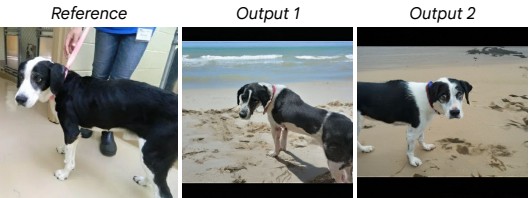

Figure 10: **Limitation.** Using only one image, the network could fail in generating the source's fine details. For example, the right eye of the dog disappears in output 1 and contains defects in output 2. **(Zoom in for best view)**

## 7  CONCLUSION

This paper raises a question of whether the dominant approach of per-object optimization for personalized image synthesis is essential, and proposes a solution for the question. We introduce a general framework of using an encoder to capture object concept so that test-time optimization can be bypassed. We then study the unique challenges in the framework. In particular, we propose a regularized joint training scheme to preserve object identity without compromising editing capability. We further propose an autocaptioning scheme to provide diverse text captions for better personalization. Our framework is able to synthesize images of the same object using texts provided by users using as few as one image in a single feed-forward pass, outperforming existing works in both quality and efficiency. We believe that the findings and insights in this work would inspire future works in improving the efficacy and applicability of personalized image synthesis approaches.

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

## A    SUBJECT EMBEDDING LAYER

We follow the model design, training and inference described in Imagen Saharia et al. (2022) unless specified. One change to the base model architecture is that to incorporate object embeddings, we add an additional cross-attention layer between the original self-attention layer and text-image cross attention layer in the transformer block, as shown in Fig. 11.

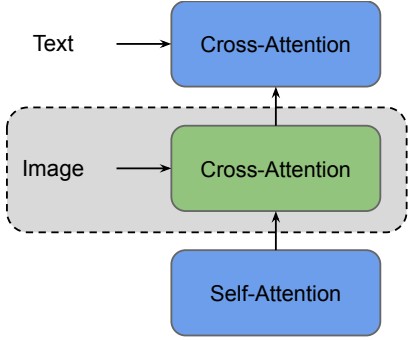

Figure 11: Newly added cross-attention layer.

As discussed in Sec 3.1, to encourage model to generalize to object embeddings and maintain the text-to-image generative prior, we design regularized joint training, including *cross-reference regularization*, *object-embedding dropping* and *whole network tuning*.

## B    MORE RESULTS

In this section, we demonstrate the capability of our method in manipulating the poses and expression of human faces. We also show additional customization results.

### B.1    POSE MANIPULATION

As shown in Fig. 12, our method is capable of generating images with various poses through specifying the angles.

### B.2    ATTRIBUTE MANIPULATION

As shown in Fig. 14 and Fig. 13, our method is also capable of altering the attributes of faces, including expression and accessories.

### B.3    CUSTOMIZATION

As shown in Fig. 15, our method is able to generate images with diverse context without altering the object identity. In particular, by fixing the text embedding and changing the object embedding, we are able to generate images with the same scene, with different objects.

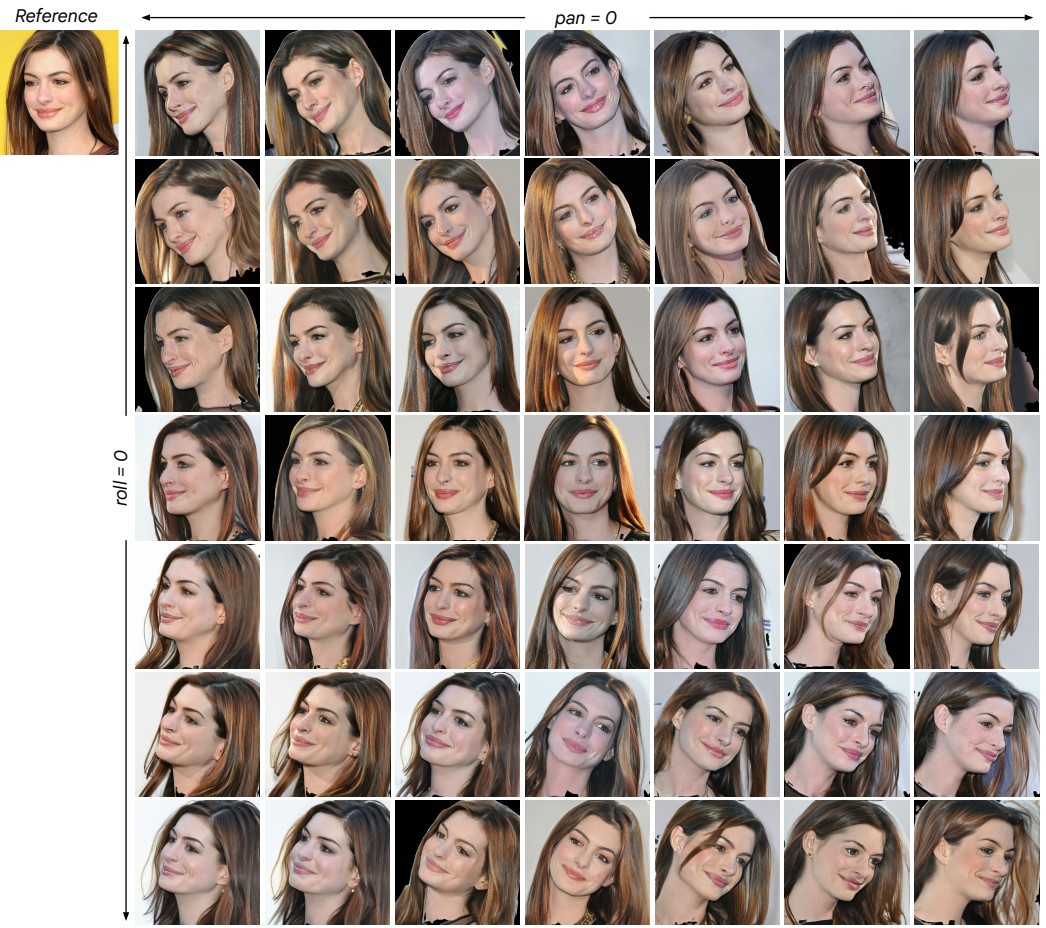

Figure 12: **Pose Manipulation.** Through specifying the angles in the text, our method is able to alter the pose of the faces.

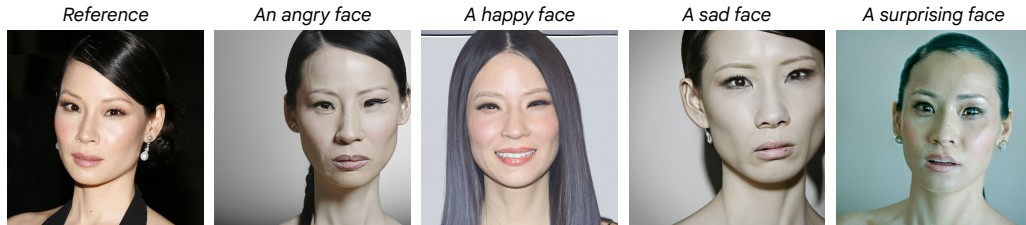

Figure 13: **Accessory Manipulation.** Our method is able to manipulate facial expression while maintaining identity.

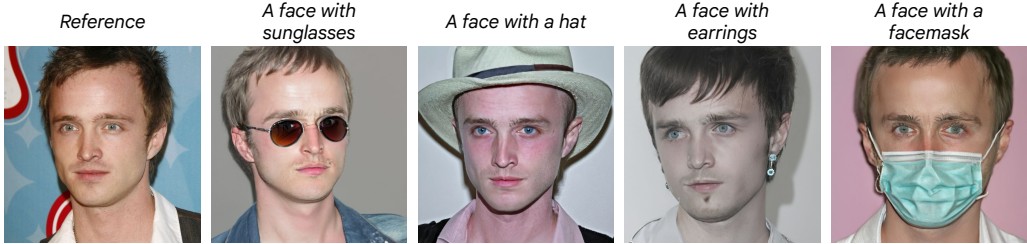

Figure 14: **Expression Manipulation.** Our method is able to manipulate accessories while maintaining identity.

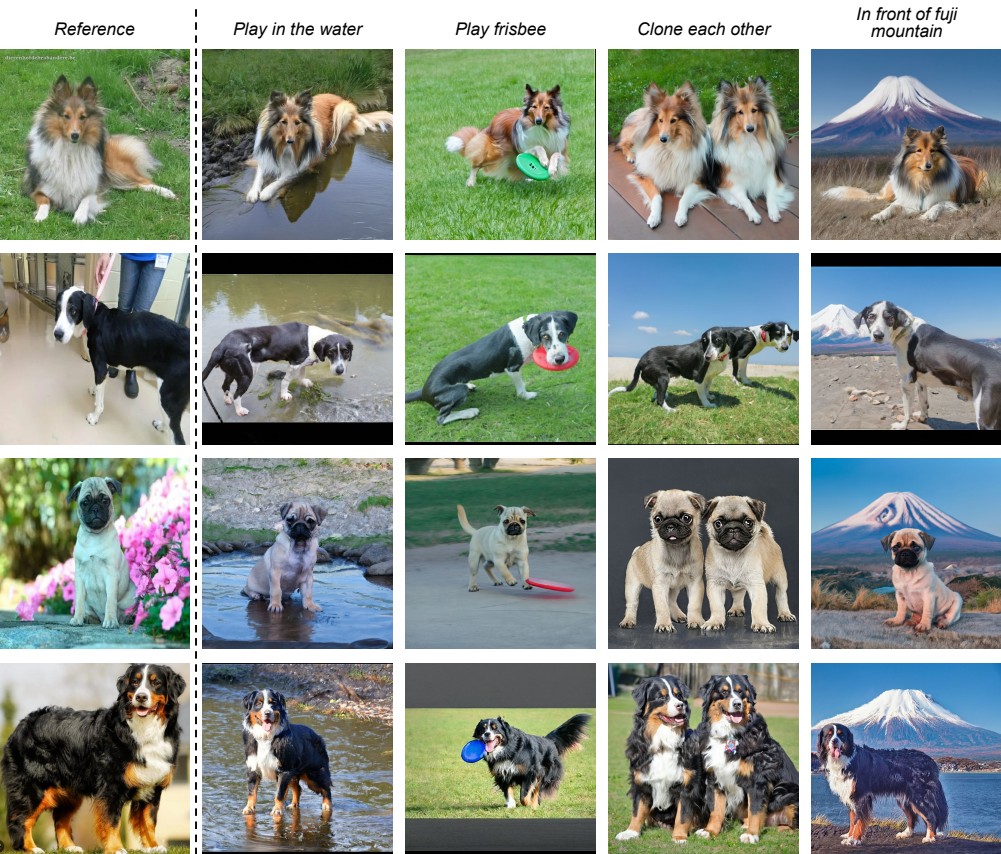

Figure 15: **More customization results on animals.** Through combining the object embedding and the text embedding, our method is able to synthesize images with diverse contexts without altering the object identity.

