# OpenReview forum: "Taming Encoder for Zero Fine-tuning Image Customization with Text-to-Image Diffusion Models"
_ICLR.cc/2024/Conference — ICLR 2024 Conference Withdrawn Submission_

### Official Review · Reviewer_un4Z · 2023-10-25

**Soundness:** 3 good
**Presentation:** 3 good
**Contribution:** 2 fair
**Rating:** 5
**Confidence:** 3

**Summary:**

This paper proposes a framework which employs an encoder to capture the essential high-level characteristics of objects, generating an object-specific embedding through a single feed-forward pass and inject the acquired object embedding to a text-to-image synthesis model for image generation. Furthermore, the paper proposes a regularized joint training scheme and a caption generation scheme, to enhance the editing flexibility and object accuracy.

**Strengths:**

1.	The paper is mostly well-organized and easy to follow.

2.	The proposed caption generation scheme is useful and applicable.

**Weaknesses:**

1. The core idea of this paper lacks innovation. Paper [1] has proposed a method of combining image embedding features and text embedding to address the time-consuming problem caused by single-image fine-tuning.

2. The paper states that the highlight of their approach is to diminish the computational burden and storage requirements. However, the paper lacks relevant comparative experiments on parameter quantity and inference time.

3. The paper lacks ablation analysis for various schemes, such as the caption generation scheme, cross-reference regularization, and object-embedding Dropping.

[1] ELITE: Encoding Visual Concepts into Textual Embeddings for Customized Text-to-Image Generation

**Questions:**

Please see the weakness.

---

### Official Review · Reviewer_QiNE · 2023-10-29

**Soundness:** 3 good
**Presentation:** 2 fair
**Contribution:** 3 good
**Rating:** 5
**Confidence:** 4

**Summary:**

This paper tackles an interesting and challenging problem: generating images of the given objects without the need for per-object optimization. The authors achieve this by introducing an object encoder to encode the identity information of the given objects. It is jointly trained with the diffusion model on domain-specific datasets and general-domain datasets. The domain-specific dataset is leveraged to encourage the disentanglement of identity information, while the general-domain dataset is leveraged to prevent the domination of the object encoder. The experiments verify the zero fine-tuning image customization of the proposed scheme.

**Strengths:**

* Generating images of the given objects without the need for per-object optimization is interesting and valuable.
* There is one illustrated insight: the object encoder is encouraged to extract the identity information instead of the whole appearance, which is implemented by cross-reference regularization.

**Weaknesses:**

* The proposed method enables customization without per-object optimization, but it seems to be limited to specific categories.
* Object-Embedding Dropping is proposed for applying the object encoder to the images in the general-domain dataset. However, it is not verified how it will perform on the general-domain data.
* The paper is not well organized, for example, it is hard to get information from Fig. 2 (right) before reading the whole text. It is also not clear whether the object encoder is initialized with the CLIP or trained from scratch, since it seems the authors fine-tune the CLIP image encoder from the description in experiments but it is not mentioned in the method.
* There is a paper that aims at the same problem [1], though it is not a peer-reviewed paper, and I will not consider it as the reason for rejection. It is better to mention it in the related works and discuss the difference.

[1] InstantBooth: Personalized Text-to-Image Generation without Test-Time Finetuning. arXiv 2023.

**Questions:**

* It is not clear that, is the object encoder trained separately for domain-specific datasets? E.g., one for dog/animal data, and one for facial data.
* For animals, are there images with the same identity? If not, how to implement the cross-reference regularization.
* Is the object encoder initialized with the CLIP or trained from scratch?
* What's the difference between customization with one image and single image editing?

**Details Of Ethics Concerns:**

The method is able to generate fake images for a specific identity, which may be abused for misinformation.

---

### Official Review · Reviewer_qmCv · 2023-10-31

**Soundness:** 3 good
**Presentation:** 3 good
**Contribution:** 3 good
**Rating:** 6
**Confidence:** 3

**Summary:**

This paper presents a novel framework for creating customized images containing user-specified objects using a single feed-forward pass, without needing per-object optimization.
It proposes a regularized joint training scheme with generated captions to maintain editing capabilities while adapting the model to object embeddings.
This approach enables efficient personalized image generation while reducing computational and storage costs compared to existing methods.

**Strengths:**

1. The paper proposes a reasonable and effective solution for generating customized images containing user-specified objects, without needing per-object optimization.
2. It identifies key challenges like the model focusing on object embeddings and forgetting text conditions, and proposes solutions like the regularized joint training scheme.
3. Extensive experiments verify the effectiveness of the proposed method across different metrics.

**Weaknesses:**

1. Some useful ablations are missing. For example, the paper proposes segmenting main objects from backgrounds, but doesn't analyze how the background of an image influences model performance.

**Questions:**

There seems to be a typo on page 9 line 6 that says "second row" but may want to refer to the "second column" of the table.

---

### Official Review · Reviewer_ddow · 2023-10-31

**Soundness:** 3 good
**Presentation:** 3 good
**Contribution:** 2 fair
**Rating:** 5
**Confidence:** 4

**Summary:**

This paper discusses a novel approach for "Personalized Text-to-Image Synthesis" that allows for the creation of customized images based on user specifications, eliminating the need for optimizing each object individually or using multiple images of the same object. This approach comprises an encoder to capture key object characteristics and a text-to-image synthesis network that generates images based on both object and text embeddings. To effectively integrate object-conditioning while preserving pre-trained model editing capabilities, a regularized joint training scheme is employed, involving techniques like cross-reference regularization, object-embedding dropping, and whole-network tuning. Additionally, a caption generation scheme is described, which enriches training data by generating descriptive captions for domain-specific images using a language-image model and an attribute classifier.

**Strengths:**

The simplicity of the architecture and the impressive results it achieves make it a promising approach for text-to-image synthesis.
The process of disentangling objects from their backgrounds in this approach offers various advantages and enhances the overall quality of image generation. Cross-reference regularization introduces a novel and effective concept for improving the integration of object-conditioning in the synthesis network. Object-Embedding Dropping is a technique within the regularized joint training scheme that helps maintain the pre-trained model's editing capabilities while incorporating object-specific information.

**Weaknesses:**

The approach bears a strong resemblance to the Encoder for Tuning (E4T) method (https://arxiv.org/pdf/2302.12228.pdf) , implying a potential connection or influence between the two techniques. Not conducting an ablation study involving masked regions is noteworthy because it is a critical element of the proposed technique, and its absence may raise questions about its effectiveness and impact on the results.

**Questions:**

The reviewer would like to see comparisons with Encoder for Tuning (E4T) method (https://arxiv.org/pdf/2302.12228.pdf). The reviewer would also like to see the ablation studies with the masking based loss.